# Clinical Outcome, Cognition, and Cerebrovascular Reactivity after Surgical Treatment for Moyamoya Vasculopathy: A Dutch Prospective, Single-Center Cohort Study

**DOI:** 10.3390/jcm11247427

**Published:** 2022-12-14

**Authors:** Pieter Thomas Deckers, Annick Kronenburg, Esther van den Berg, Monique M. van Schooneveld, Evert-Jan P. A. Vonken, Willem M. Otte, Bart N. M. van Berckel, Maqsood Yaqub, Catharina J. M. Klijn, Albert van der Zwan, Kees P. J. Braun

**Affiliations:** 1Department of Neurology and Neurosurgery, UMC Utrecht Brain Center, 3584 CG Utrecht, The Netherlands; 2Department of Radiology and Nuclear Medicine, Meander Medisch Centrum, 3813 TZ Amersfoort, The Netherlands; 3Department of Neurology, Erasmus University Medical Center, 3015 GD Rotterdam, The Netherlands; 4Department of Pediatric Psychology, Wilhelmina Children’s Hospital, 3584 EA Utrecht, The Netherlands; 5Department of Radiology, UMC Utrecht Imaging Division, 3508 GA Utrecht, The Netherlands; 6Department of Nuclear Medicine & PET Research, Amsterdam UMC, 1081 HV Amsterdam, The Netherlands; 7Department of Neurology, Donders Institute for Brain, Cognition and Behavior, Center for Neuroscience, Radboud University Medical Center, 6525 GA Nijmegen, The Netherlands

**Keywords:** moyamoya disease, cerebral revascularization, cognition, cerebrovascular reactivity, ischemia, quality of life

## Abstract

Background: It remains unclear whether revascularization of moyamoya vasculopathy (MMV) has a positive effect on cognitive function. In this prospective, single-center study, we investigated the effect of revascularization on cognitive function in patients with MMV. We report clinical and radiological outcome parameters and the associations between clinical determinants and change in neurocognitive functioning. Methods: We consecutively included all MMV patients at a Dutch tertiary referral hospital who underwent pre- and postoperative standardized neuropsychological evaluation, [^15^O]H_2_O-PET (including cerebrovascular reactivity (CVR)), MRI, cerebral angiography, and completed standardized questionnaires on clinical outcome and quality of life (QOL). To explore the association between patient characteristics, imaging findings, and change in the *z*-scores of the cognitive domains, we used multivariable linear- and Bayesian regression analysis. Results: We included 40 patients of whom 35 (27 females, 21 children) were treated surgically. One patient died after surgery, and two withdrew from the study. TIA- and headache frequency and modified Rankin scale (mRS) improved (resp. *p* = 0.001, 0.019, 0.039). Eleven patients (seven children) developed a new infarct during follow-up (31%), five of which were symptomatic. CVR-scores improved significantly (*p* < 0.0005). The language domain improved (*p* = 0.029); other domains remained stable. In adults, there was an improvement in QOL. We could not find an association between change in imaging and cognitive scores. Conclusion: In this cohort of Western MMV patients, TIA frequency, headache, CVR, and mRS improved significantly after revascularization. The language domain significantly improved, while others remained stable. We could not find an association between changes in CVR and cognitive scores.

## 1. Introduction

Moyamoya vasculopathy (MMV) is a cerebrovascular disorder of largely unknown etiology, characterized by progressive stenosis or occlusion of the supraclinoid internal carotid arteries and their proximal branches [1,2]. Most patients present with transient ischemic attacks (TIAs) or ischemic or hemorrhagic stroke and others with cognitive impairment [3]. Idiopathic MMV is referred to as moyamoya disease (MMD); MMV associated with another predisposing condition, e.g., neurofibromatosis, is referred to as moyamoya syndrome (MMS) [1]. Revascularization surgery is recommended for patients with ischemic symptoms or disturbed cerebrovascular reactivity (CVR) [2,4,5]. To identify brain areas at risk of ischemia, [^15^O]H_2_O-positron emission tomography (PET) is commonly used and enables the assessment of CVR after acetazolamide challenge [2].

We recently studied the cognitive profile of 40 MMV patients and found that 73% had cognitive impairments in at least one domain; children performed better in processing speed, and adults had higher scores in visuospatial functioning [6]. Little is known about the effect of revascularization on cognition in MMV, especially in the Western world [3], and on quality of life (QOL) [7].

In this prospective cohort study, we investigated the effect of revascularization on clinical outcome and cognition in patients with MMV and to what extent cognitive functions are related to CVR.

## 2. Materials and Methods

### 2.1. Patient Selection

The Medical Ethics Review Committee UMC Utrecht confirmed that the Medical Research Involving Human Subjects Act (WMO) did not apply. All patients or representatives gave written informed consent. We prospectively included forty consecutive MMV patients (children up to 18 years old and adults) who presented between October 2012 and September 2017 in our center, were not previously treated with revascularization surgery, could understand Dutch, were available for follow-up, and had a confirmed diagnosis of MMV by digital subtraction angiogram (DSA) or magnetic resonance angiography (MRA) [4]. Their baseline characteristics have been reported previously [6]. There were no eligible patients who refused participation.

Thirty-five patients (21 children) were treated operatively and five conservatively. Only the surgically treated group is included in the analysis. Since the treatment was tailored and not randomized, outcomes of conservatively treated patients are reported in the appendix (Appendix I and Appendix J) and were not directly compared to surgical patients.

### 2.2. Treatment

All patients took acetylsalicylic acid (38 mg for children, 100 mg for adults). Timing, type, and location of revascularization was decided in a multidisciplinary meeting by the treating neurologist and neurosurgeon based on symptoms, [^15^O]H_2_O-PET findings, and shared decision making with the patients, parents, or caregivers. The preferred initial surgical treatment was a single-staged combined direct and indirect revascularization. In children, an additional bifrontal revascularization was performed if indicated (Appendix A for details) [6,8].

### 2.3. Clinical Follow-Up

All patients were seen for follow-up at least one year after the last operation or at least one year after the baseline visit for patients treated conservatively. Patients were neurologically examined and interviewed using standardized questionnaires and a predefined case record form (for characteristics see Appendix B).

Any peri-operative complication or adverse event < 30 days after surgery was carefully noted and reviewed at the routine 6-week postoperative outpatient evaluation. Any complications resulting in permanent deficits or requiring additional surgery were classified as “serious”, while others were classified as “transient”. In case of withdrawal or death of patients during follow-up, all available endpoints were used (e.g., death, new infarction), but these patients were excluded from the final cognitive analysis.

To assess stroke severity, we applied the Pediatric National Institutes of Health Stroke scale (pedNIHSS) [9] in children and the NIHSS [10] in adults. The modified Rankin scale (mRS) [11] was applied in both groups to assess functional status; additionally, for children, the Pediatric Stroke Outcome Measure (PSOM) [12] was used. To measure QOL, we applied the Pediatric Quality of Life Inventory in children, including the parent-proxy questionnaire (PedsQL) [13]. Adults completed the Short Form—36 (SF-36) [14] and the EuroQol EQ-5D-3L (EQ-5D) [15].

### 2.4. Imaging

Patients were evaluated by MRI, digital subtraction angiography (DSA), and [^15^O]H_2_O-PET in combination with an acetazolamide challenge to determine CVR, as described previously (see also Appendix C for further details) [6].

MRI and [^15^O]H_2_O-PET images were scored in three global regions of interest (ROIs) of comparable size in each hemisphere (labeled as “frontal”, “middle”, and “posterior”) by two reviewers (P.T.D. and A.K.), blinded for all patient characteristics [6,16].

The ROIs in the MRI-FLAIR were scored on the presence of infarcts using the following scoring: 0 = none; 1 = small; 2 = intermediate; and 3 = large infarct. Furthermore, we scored periventricular (WMDp) and deep white matter disease (WMDd), applying an adapted Fazekas score per ROI: 0 = none; 1 = small or subtle; 2 = intermediate; and 3 = extensive [6,17]. When the differentiation between a new white matter lesion and a small lacunar infarction was ambiguous, diffusion-weighted imaging (DWI) and T1 images were additionally reviewed.

The [^15^O]H_2_O-PET CVR was qualitatively scored per ROI using the following score: 0 = CVR normal; 1 = minimal CVR present; 2 = CVR absent; and 3 = a steal phenomenon is present in any region within the ROI (i.e., the reduction of CBF after acetazolamide administration). If the entire ROI could not be scored due to infarction, this was noted as a missing value. Furthermore, the CVR was visually compared between baseline and follow-up to rate the CVR qualitatively as “improved”, “stable”, or “deteriorated”. For the calculation of the WMD, infarctions, and CVR scores, we averaged the valid ROI scores from the six regions.

DSA images were reviewed by an experienced neuroradiologist (E.J.V.) blinded for other data. The involvement of the ACA, MCA, and PCA was assessed according to the following categories: 0 = no evidence of disease; 1 = mild to moderate stenosis with absent or slightly developed MMD collaterals; 2 = severe stenosis with well-developed MMD collaterals; 3 = occlusion with well-developed MMD collaterals; and 4 = occlusion with absent or slightly developed MMD collaterals. To categorize the overall severity of MMV, the modified Suzuki score (mSS) was determined based on the highest hemispheric score [18]. Furthermore, the change relative to the baseline DSA was noted, including increase of collaterals, bypass patency, and change in mSS.

### 2.5. Cognitive Evaluation

All patients underwent a standardized neuropsychological evaluation test battery specified for MMV (Appendix D, Table A1) at baseline and at follow-up [6]. Neurocognitive tests were planned just before the follow-up visit, after a median of sixty weeks following last surgery (range: 40–108).

Tests were specifically chosen for children or adults but covered the same predefined cognitive test domains: general functioning; memory; working memory; language; attention and executive functioning; processing speed; and visuospatial functioning [19]. We assessed cognitive domains of all patients combined and for children and adults separately. As is common in clinical studies, patients were tested according to their age and capabilities. Not all patients performed the complete neuropsychological test battery. All available data were used as an estimation of individual cognitive domains. Raw test scores were corrected for age and education level using their respective manuals. All available adjusted scores were then converted to *z*-scores and averaged per domain. Cognitive impairment was defined as 1.5 *SD* or more below the population mean (i.e., *z*-score of <−1.5) in one or more domains [6]. Two children in whom we could not establish any reliable test score due to a low developmental age or insufficient understanding of the test were assumed to have a cognitive impairment but could not be quantitatively analyzed. When deemed necessary by the treating physicians, patients received speech therapy during the follow-up.

### 2.6. Data-Analysis and Statistics

All questionnaires—(ped)NIHSS, PSOM, mRS, PedsQL, SF-36, and EQ-5D-3L—were analyzed according to their respective manuals [6]. Changes in continuous variables (neurocognitive *z*-scores, PedsQL, SF-36, and EQ-5D-3L) were analyzed using Student’s *t*-tests after visually checking normality using histograms and *q-q* plots and formally with Kolmogorov–Smirnov tests. Changes between baseline and follow-up for ordinal variables (mRS, stratified TIA and headache frequency, (ped)NIHSS, and PSOM) were analyzed using paired-samples sign tests. A *p*-value < 0.05 was considered significant.

To investigate the effect of possibly relevant determinants on postoperative change in neurocognitive functioning, we used both multivariate linear regression and Bayesian regression. These determinants included: age categories (adult versus child); change in mSS, infarction, WMD, and CVR-score; and the presence of an associated predisposing condition that could have led to cognitive deficits in children with MMS (Down or Noonan syndrome; microcephalic osteodysplastic primordial dwarfism—II (MOPD-II); neurofibromatosis(NF)-1; and posterior fossa brain malformations, hemangioma, arterial lesions, cardiac abnormalities, and eye abnormalities syndrome (PHACES)) [6]. For the Bayesian regression, the Bayes factor (*BF*) was determined. A commonly used list divides the evidence in favor of an association into four strength ranges: *BF*s 1–3.2: “not worth more than a bare mention”; 3.2–10: “substantial”; 10–100: “strong”; and >100: “decisive” evidence [20]. Furthermore, we used univariate linear regression analysis and Bayesian regression to correlate postoperative change in cognitive domain scores with baseline CVR scores and change in CVR scores for each of the three ROIs separately. Statistics were performed using SPSS Statistics version 26 (IBM Corp, Armonk, NY, USA) and Bayesian and linear regression with JASP version 0.14.0 (JASP Team (2020)).

## 3. Results

### 3.1. Patient Characteristics and Treatment

Of the 40 included patients at baseline, 35 were operated (27 females; 21 children). We excluded three adults from the analysis of cognitive outcomes (one died two days postoperatively; two withdrew from follow-up) but included their available clinical end-points (e.g., complications, MRS). The remaining 32 patients underwent MRI and [^15^O]H_2_O-PET at follow-up; 30 received a DSA (one MOPD-II child was too small; in one child, it was postponed due to perioperative infarction). Thirty-one underwent neuropsychological evaluation (the parents of one child refused follow-up evaluation), and median time of neuropsychological evaluation was 60 weeks (range 40–108).

Twenty-one patients were treated bilaterally and fourteen unilaterally (Table 1). Two of the bilaterally treated patients had a single-stage procedure: one child with an indirect bilateral fronto-parietal bypass and another child with a unilateral combined bypass and bifrontal indirect procedure. The other bilateral procedures were two-staged. Twenty-nine patients (fifteen children) received a direct bypass in at least one hemisphere.

### 3.2. Surgical Complications

Four patients (11% of 35 patients; 7.4% of 54 operations) had serious complications resulting in permanent deficits (three patients) or death (one patient) within 30 days after operation. One patient died due to an intraparenchymal hemorrhage, most probably caused by hemorrhagic transformation of an infarcted area caused by hyperperfusion. The deceased patient was in poor clinical condition pre-operatively (see Appendix E). Three patients had new permanent deficits due to infarction (two adults). One of these adults also developed a bone flap infection requiring additional surgery. The infarction led to a deterioration in mRS at follow-up one year after operation from 1 to 2 in the child. The two adults withdrew from follow-up so their follow-up mRS is unavailable. Five patients had transient deficits due to hyperperfusion syndrome, and three patients had TIAs during the first 30 days following surgery. Further details are provided in Appendix E.

### 3.3. Clinical Follow-Up

There were 26 patients who had preoperative TIAs. Of those 26 patients the TIA frequency improved after surgery in 19 (73%); 16 became completely TIA-free (62%, 11 children, Figure 1A); and in 4, frequency remained unchanged. Overall TIA frequency improved significantly for the total group (*p* = 0.001) and for children (*p* < 0.0005), while there was no significant change for adults (*p* = 0.727).

### 3.4. Headache

Twenty-seven patients presented with headache (fifteen children, Figure 1C), of whom twelve had weekly complaints. Postoperatively, frequency improved in fifteen patients (43%; eight children), remained stable in thirteen (37%; nine children) and deteriorated in four patients (11%; all children). Frequency improved significantly in the total group (*p* = 0.019) and in adults (*p* = 0.016), not in children (*p* = 0.398).

### 3.5. Other Clinical Outcomes

Three patients (two children) presented with seizures: two became asymptomatic, and one remained stable. One child presented with chorea, which completely resolved postoperatively.

### 3.6. mRS, (ped)NIHSS, PSOM

MRS (Figure 1B) at follow-up improved in ten patients (29%, seven children) and deteriorated in three (9%; one child). Overall scores improved significantly for the total group (*p* = 0.039) and for children (*p* = 0.039; Appendix F, Table A2). Median pedNIHSS was 0 (range 0–3) at baseline and 0 (0–2) at follow-up, NIHSS was 1 (0–7) at baseline and 1 (0–2) at follow-up. Median PSOM at baseline was 0 (0–3) and 0 (0–2) at follow-up. These changes were non-significant.

### 3.7. Imaging

Eleven patients (31%, seven children) developed new infarctions on MRI (Figure 2A). Five (two children) were symptomatic, and in three patients, infarcts occurred peri-operatively. The average infarction score for the total group at follow-up increased from 0.68 to 0.82 (95%*CI* −0.27–−0.018, *p* = 0.027). Ten patients had an increase in WMD score (29%, eight children, Figure 2B); the average WMD score for the total group increased from 0.61 to 0.66 (95%*CI* −0.103–0.013, *p* = 0.127).

DSA showed an increase of extracranial-to-intracranial collaterals in all operated hemispheres. In patients who underwent direct bypass surgery (*n* = 29), all bypasses were open. The mSS improved in two patients (one child, Figure 2C) and deteriorated in five (three children). Mean mSS remained stable (3.0 at baseline, 3.1 at follow-up (95%*CI*: −0.279–0.079, *p* = 0.264)).

Average CVR scores improved in 24 patients (15 children, Figure 2D and Figure 3), deteriorated in 5 (three children), and remained stable in 2 (both children). Group CVR scores improved from 1.93 at baseline to 0.82 at follow-up (95%*CI*: 0.668–1.567, *p* ≤ 0.0005).

### 3.8. Neuropsychological Evaluation

Cognitive functioning on a group level at follow-up was comparable with baseline [6]: postoperative test scores of all domains were significantly lower than the population mean in the total group and in children except for attention and executive functioning (Figure 4A) and visuospatial functioning in adults.

When looking at the change in *z*-scores in the total group, language domain scores improved significantly (*p* = 0.029), while the other mean *z*-scores remained stable (Figure 4B). In adults, none of the domain scores significantly changed after surgery, while children showed a significant improvement in language domain functioning (*p* = 0.006). Since the *z*-scores are corrected for age, this improvement exceeds the expected normal development.

There were no significant changes in the number of patients with cognitive deficits after surgery. Seven patients (six children) had one or more deficits in a domain at baseline that resolved after follow-up (Figure 4A). However, in four patients (three children), a new deficit developed in another domain. One child improved from having cognitive impairment to none at follow-up; one adult developed cognitive impairment postoperatively; the others remained stable. To show the variability of changes between different cognitive domain scores in single patients, individual changes are graphically depicted in Appendix G, Figure A1.

### 3.9. Quality of Life

In children, QOL scores remained stable: the summary, PedsQl scores went from 74.1 to 76.7 (95%*CI*: −18.8–13.3, *p* = 0.73) and the parent-proxy score from 70.2 to 69.4 (95%*CI*: −7.0–8.4, *p* = 0.85). In adults, the mean of the physical component summary of the SF-36 remained stable from 47.0 to 50.1 (95%*CI*: −8.3–1.99, *p* = 0.19), and the mental component summary improved from 44.4 to 50.3 (95%*CI*: −10.7–−1.1, *p* = 0.022). The visual analogue scale of the EQ-5D improved from 73.9 to 82.0 (95%*CI*: −14.2–−2.99, *p* = 0.015), and the index remained stable (0.880 to 0.862 (95%*CI*: −0.090–0.1262, *p* = 0.715)).

### 3.10. Correlation of Imaging Changes to Neurocognitive Changes

We tested the hypothesis that changes in imaging parameters and CVR were associated with a change in cognitive scores (Table 2). Improvement in mSS was significantly associated with improvement in memory (*p* = 0.019). Children improved significantly more than adults in the working memory domain (*p* = 0.027). Finally, a worse score of WMD correlated to an improved visuospatial functioning score. We found no significant association between changes in clinical and radiological parameters (especially CVR) and changes in the other cognitive domains.

When comparing baseline CVR in each of the three ROIs (frontal, middle, and posterior) separately to postoperative changes in cognition in a univariable regression analysis, we found a significant correlation between worse baseline CVR scores in the middle and posterior regions and improvement in the cognitive domain visuo-spatial functioning (*B* = −0.378 (95%*CI*: −0.634–−0.122, *p* = 0.006); *B* = −0.314 (95%*CI*: −0.569–−0.059, *p* = 0.018), respectively; Appendix H, Table A3). All other associations were non-significant.

## 4. Discussion

In this prospective, single-center Dutch cohort study of 35 operatively treated MMV patients, we showed a significant improvement in frequency of TIAs and headache and mRS after revascularization. CVR improved significantly in both children and adults. Neuropsychological evaluation showed that patients performed significantly beneath population mean on all domains except for attention and executive functioning on a group level. After revascularization, language improved significantly—in the total group and in children—whereas other domains remained stable. While the improvement was corrected for age, it might be influenced by the speech therapy some of the patients received. QOL remained stable in children, while we found significant improvement in adults. We found no statistically significant associations between changes in clinical, radiological, and hemodynamic variables and cognitive domain scores.

Cognitive outcome following surgery may be expected to differ between children and adults. Although cognitive scores take age-specific reference values into account, and the use of *z*-scores allows for pooling of test results, inherent differences between age groups justified the analysis and presentation of results for children and adults separately. The eventual postoperative cognitive domain *z*-scores showed a relative vulnerability for visuospatial functioning in children and for processing speed in adults. Multivariable change of cognitive scores was not significantly determinant by age group except for the working memory domain. The single domain that showed postoperative significant improvement—and only so in children—was language. Possibly, the younger brain of children with a cerebrovascular compromise has a higher potential of functional recovery and improved language development than that of adults.

We saw eight new infarctions not associated with the surgical treatment between baseline and follow-up (median follow-up time: 21 months, Table 1). Only two of these led to clear clinical symptoms. In the conservative group, we saw no new ischemic lesions on follow-up MRI (Appendix I). Since the treatment was not randomized but specifically tailored to the patient, this group is inherently different from the surgical patients, so they cannot be directly compared. Furthermore, follow-up duration was shorter in the conservative group, making an ischemic event less likely to occur.

The compromised cerebrovascular hemodynamics and fragile MMV vessels may lead to a high surgical risk [21]. Therefore, patients need to be carefully selected for surgical treatment, and maintaining adequate blood pressure during anesthesia is of great importance. Even with precautionary measures, the risk of complications in our cohort was high, with four patients (11%) experiencing severe complications, of whom one, who was in a poor preoperative condition, died.

Since prospectively performed studies are rare, the overall mortality rate of surgical treatment of MMV remains unclear.

Previous studies reported mortality ranges between 0.86% [22] and 1.86% [23]. Overall adverse postoperative events (mainly ischemic and hemorrhagic stroke) are reported to be between 5–14% [22,23,24] in MMV, consistent with our results. Remarkably, the only randomized controlled surgical trial in MMD comparing STA-MCA bypass surgery to conservative treatment in adults who presented with hemorrhage reported not a single perioperative adverse event after 84 operations [25]. In another study, postoperative routine DWI revealed new ischemic lesions in 9% of 140 procedures [26]. In the subgroup of twenty-four procedures in patients who were considered to have “unstable MMD” (defined as rapid stenosis progression or recurrent stroke), 33% had postoperative DWI lesions, suggesting that postoperative ischemia is not uncommon [26]. In our study, imaging was not routinely performed directly postoperatively. Therefore, it remains unknown what proportion of silent infarctions was associated with surgery.

Our finding that revascularization surgery reduces TIA frequency in children is consistent with other studies in adults [27] and children [28]. The beneficial effect of revascularization on headache has also been reported before in children [29] and adults [30] although headache as a primary outcome is probably underreported.

Revascularization surgery has been shown to improve CVR [31]. This is confirmed by our study. Impaired CVR has previously been linked to cognitive decline [32,33]. We could not demonstrate a direct correlation between improvement of CVR and cognitive improvement. Several studies have investigated the relationship between cerebral hemodynamics and cognition—as described below—but consistent associations were not found. Asian and Western MMV populations appear to differ, with Asian MMV patients tending to be younger, presenting more often with hemorrhages, and being often more severely affected. Therefore, associations between CVR and cognition are possibly not directly comparable between populations [34].

One Western study showed that—although there was no improvement in CVR—there was some improvement in executive functioning [24]. The difference in CVR change could be explained by the burr-hole technique used, possibly leading to a slower increase in hemodynamic functioning than the combined direct–indirect technique we used. Another study showed that postoperative cognitive function remained stable in 75% of patients, significantly deteriorated in 14%, and improved in 11% of the patients [35]. These results are in line with ours. Although improvement of cognitive functioning would ideally be aimed for, in a progressive disease such as MMV, stabilization of cognition may still be considered a positive outcome. Furthermore, cognitive decline in conservatively treated MMS patients has been previously shown [36]. We hypothesize that revascularization prevents this decline in selected patients.

A Japanese prospective study showed improvement in cognitive functioning in adults after bilateral direct revascularization, with some cognitive tests correlating to preoperative CBF [37]. However, the increase was only visible two years after treatment, which could explain the difference with our results. In addition, IQ scores in those studies were—in contrast to ours—within normal range at baseline. Another retrospective cohort study found no significant difference between pre- and postoperative IQ, while the cerebral perfusion improved in several regions [38]. Change in perfusion did not correlate with change in cognitive tests. However, a high baseline oxygen ejection fraction (OEF) correlated with improvement in performance IQ, and improvement of cerebral metabolic rate of oxygen correlated with improvement in verbal IQ. Another prospective study in conservatively treated MMD patients without misery perfusion on PET showed that the CBF increased significantly although no significant cognitive changes were observed [39]. This study, however, entailed a selected group since only mildly affected MMD patients show no misery perfusion. Overall, the results of these studies are similar to ours although we used different hemodynamic parameters.

Only a few retrospective Asian studies have assessed the correlation between cognitive and cerebrovascular changes exclusively in children. One showed that CBF increased in all hemispheres, and in some ROIs, this correlated with change in cognition [40]. However, there were also territories with a negative correlation between CBF and cognitive test results, making it more difficult to draw definite conclusions. Two other studies suggested some improvement of cognition after treatment, but this was not correlated to CVR change [41,42].

QOL in MMV is underreported [7]. We showed that the PedsQL remained stable after treatment in children. Our results align with norm scores for Dutch children with a chronic health condition [43] and with another study of surgically treated children with MMD [7]. A study from the U.K. using only the parent-proxy of the PedsQl reported a lower score of 66.0 compared to the children’s scores in our cohort [44]. In adults, we found a statistically significant improvement for the QOL as measured by the VAS from the EQ-5D and the MCS of the SF36 but not for the EQ-5D index nor for the PCS. After treatment, the VAS scores were close to the population norms [45].

An important limitation of our study—and one of the possible reasons for not being able to demonstrate a correlation between surgery, change in hemodynamic measures, and cognitive outcome—is the relatively short follow-up time. It might take longer to show quantifiable cognitive improvement than just one year [37]. Furthermore, our cohort included the full range of MMS/MMD patients, making it a heterogeneous group possibly distorting the outcome. Next, the cohort was too small for further subgroup analysis. Heterogeneity of the cohort also implied that some patients had such low cognitive performance that they could not complete the complete test battery, making the reported *z*-scores an overestimation of the groups’ cognitive abilities. Furthermore, the [^15^O]H_2_O-PET was not fully quantitated, and the ROI-based analysis was not accurate enough to specifically look into specific anatomical regions, possibly affecting the sensitivity of our analysis. One of the strengths of this study is its prospective design, resulting in a detailed diagnostic follow up evaluation. We specifically choose to include pediatric, adult, and MMS and MMD patients, reflecting standard clinical practice. MMV is a rare disease, and this is one of the largest Western MMV cohorts with detailed cognitive test results available to date.

Despite our research and all previous efforts, there are still unanswered questions regarding MMV treatment. It remains unknown if the patients we chose to operate would have improved as much without surgery, nor do we know the best revascularization method if surgery is indeed indicated. More research is needed to understand how QOL in MMV can be improved. For tailored treatment strategies in different subgroups (e.g., children/adults, MMD/MMS, primary presentation of hemorrhage or ischemia) with the optimal revascularization method, future research should focus on standardized, multicenter prospective studies to improve knowledge on treatment of MMV.

## 5. Conclusions

In this prospective, single-center cohort study of MMV patients, we showed that one year after revascularization, CVR improved, and cognition remained stable in most domains and significantly improved in the language domain, specifically in children. Furthermore, TIA frequency and mRS improved significantly in children, while we found improvements in headache and QOL in adults. We could not find a relationship between change in CVR and change in neurocognitive parameters. We report a rate of serious complication of 11%, which is high but comparable to what was previously reported, stressing the importance of carefully counseling the patients of risk involved before surgery.

## Figures and Tables

**Figure 1 jcm-11-07427-f001:**
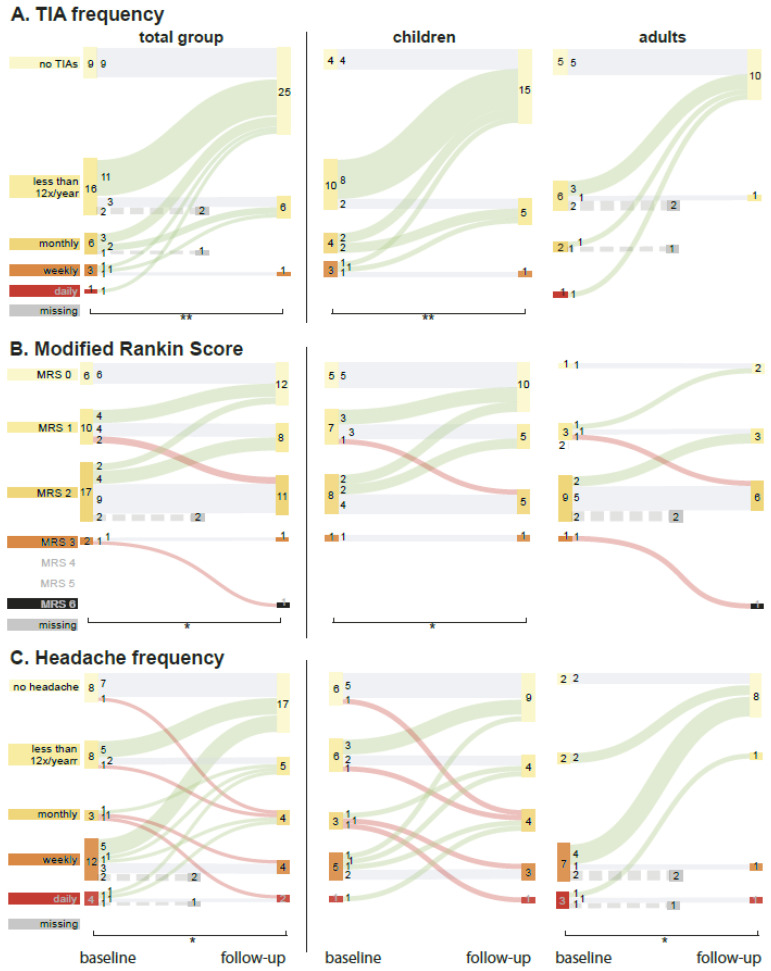
Clinical outcome of treated patients for the total group and for children and adults separately. The width of the bars represents the number of patients and is described by the value within. Green upward line = improvement; horizontal grey line = stable; downward red line = deterioration. Dotted line represents a patient with missing outcome data. (**A**) Effect of operative treatment on TIA frequency. (**B**) Change in postoperative modified Rankin scale (mRS). (**C**). Headache frequency before and after treatment. Significant differences between baseline and follow-up are denoted with a * (*p*-value of <0.05) and ** (*p* < 0.005).

**Figure 2 jcm-11-07427-f002:**
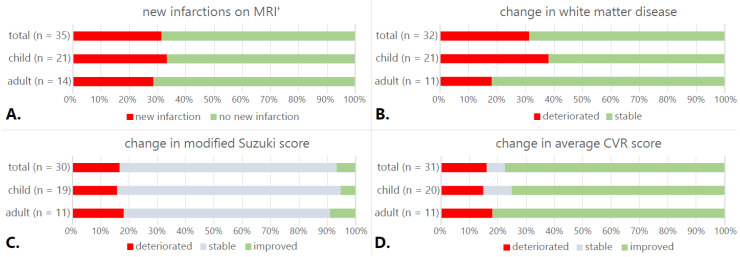
Change in imaging scores. (**A**) New infarctions on the MRI one year after the last operation. (†) Three adults did not receive a scan at follow-up but received an MRI shortly after the operation and are included in this graph. (**B**) Change in average white matter disease. (**C**) Change in modified Suzuki score (mSS). (**D**) Change in average cerebrovascular reactivity (CVR) score.

**Figure 3 jcm-11-07427-f003:**
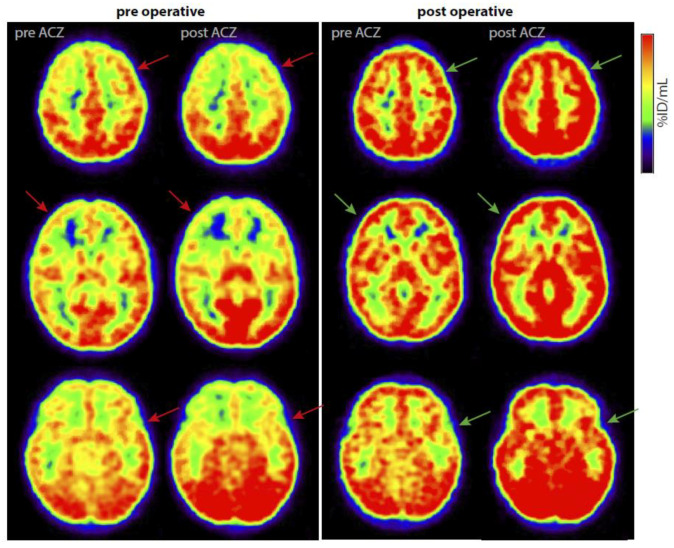
Example of improvement of cerebrovascular reactivity (CVR) on [^15^O]H_2_O-PET in an 11-year-old girl with moyamoya syndrome. The left two columns are pre-operative and before and after acetazolamide (ACZ) challenge. The red arrows show examples of vascular steal. She was first operated with a right-sided combined direct/indirect bypass with additionally a bifrontal EDAMS, followed by a left-sided indirect bypass during a second surgery. The right two columns show the same patient approximately at follow-up. The green arrows show the same areas with improved CVR.

**Figure 4 jcm-11-07427-f004:**
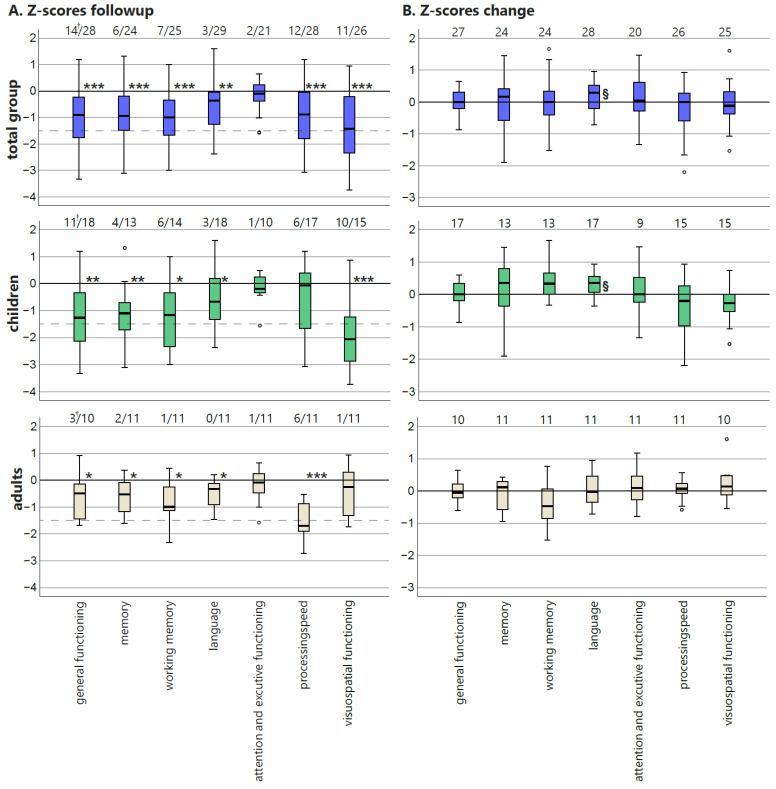
(**A**) *Z*-scores of neurocognitive tests at follow-up for the total group for children and adults separately for all seven cognitive domains. The numbers represent the amount of patients with a *z*-score below −1.5 *SD* (dashed line) compared to the total number of patients with a valid domain score per group. Scores significantly different from the population mean (one-sample *t*-test) are noted with * (*p* < 0.05), ** (*p* < 0.005), and *** (*p* < 0.0005). (†) Two pediatric patients could not understand the tests due to insufficient cognitive functioning. These patients were given a “deficit” score for general functioning but could not be included in the box plots. (**B**) Change in *z*-score between follow-up and baseline. A higher score indicates an improvement. The numbers represent the amount of valid test scores per cognitive domain; (^§^) significant improvements in *z*-score (*p* ≤ 0.05).

**Table 1 jcm-11-07427-t001:** Patient characteristics and treatment.

		Total Group (*n* = 35)		Children (*n* = 21)		Adults (*n* = 14)	
Sex	Female	27	77.10%	15	71.40%	12	85.70%
Moyamoya diagnosis	MMD unilateral	1	2.90%	0	0.00%	1	7.10%
	MMD bilateral	22	62.90%	11	52.40%	11	78.60%
	MMS unilateral	3	8.60%	3	14.30%	0	0.00%
	MMS bilateral	9	25.70%	7	33.30%	2	14.30%
Age at follow-up in years (mean (*SD*))		20.6 (15.5)		11.0 (4.1)		38.8 (12.4)	
Time between baseline and follow-up in months (median (min-max))		21 (14–72)		21 (14–72)		20 (14–24)	
Time between last OR and follow-up in months (median (min–max))		15 (10–27)		16 (10–27)		12 (10–24)	
Treatment type	Bilateral treatment	21	60.00%	16	76.20%	5	35.70%
	Unilateral treatment	14	40.00%	5	23.80%	9	64.30%
	Total patients with direct bypass	29	82.90%	15	71.40%	14	100.00%
	Total patients with frontal procedure	13	37.10%	13	61.90%	0	0.00%

All operated patients are included in this table. MMD, moyamoya disease; MMS, moyamoya syndrome; *SD*, standard deviation; OR, operation.

**Table 2 jcm-11-07427-t002:** Multivariable regression analysis of possible determinants of change in cognitive domain scores.

**General Functioning**	*B (95%CI)*	*p*	*BF*
Adult vs. child	0.14 (−0.345–0.624)	0.552	0.325
Infarct score change	−0.103 (−0.996–0.791)	0.811	0.316
MSS change	−0.045 (−0.504–0.414)	0.837	0.295
WMD score change	0.942 (−0.641–2.525)	0.226	0.469
CVR score change	−0.053 (−0.267–0.161)	0.610	0.314
Other reason for cognitive defect	0.307 (−0.399–1.013)	0.371	0.361
**Memory**	*B (95%CI)*	*p*	*BF*
Adult vs. child	−0.606 (−1.469–0.258)	0.155	0.813
Infarct score change	1.326 (−0.583–3.236)	0.158	0.854
mSS change	−0.963 (−1.741–−0.185)	**0.019**	2.474
WMD score change	−0.972 (−4.942–2.997)	0.608	0.608
CVR score change	−0.020 (−0.368–0.329)	0.905	0.571
Other reason for cognitive defect	0.144 (−1.821–2.11)	0.877	0.630
**Working Memory**	*B (95%CI)*	*p*	*BF*
Adult vs. child	−1.057 (−1.974–−0.140)	**0.027**	1.965
Infarct score change	0.294 (−1.733–2.322)	0.760	0.51
mSS change	0.465 (−0.360–1.291)	0.247	0.599
WMD score change	−0.609 (−4.823–3.605)	0.761	0.505
CVR score change	−0.191 (−0.561–0.179)	0.286	0.585
Other reason for cognitive defect	−0.652 (−2.739–1.435)	0.514	0.483
**Language**	*B (95%CI)*	*p*	*BF*
Adult vs. child	−0.330 (−0.788–0.129)	0.148	0.682
Infarct score change	0.11 (−0.753–0.974)	0.791	0.360
mSS change	−0.347 (−0.791–0.096)	0.117	0.686
WMD score change	−0.110 (−1.640–1.421)	0.882	0.356
CVR score change	0.035 (−0.172–0.242)	0.727	0.375
Other reason for cognitive defect	−0.095 (−0.778–0.587)	0.772	0.366
**Attention and Executive Functioning**	*B (95%CI)*	*p*	*BF*
Adult vs. child	0.172 (−0.710–1.055)	0.676	0.528
Infarct score change	−1.706 (−3.722–0.31)	0.090	1.214
mSS change	−0.186 (−1.044–0.671)	0.642	0.526
WMD score change	0.678 (−3.322–4.678)	0.716	0.655
CVR score change	−0.144 (−0.496–0.208)	0.387	0.658
Other reason for cognitive defect	1.338 (−0.519–3.196)	0.141	0.716
**Processing Speed**	*B (95%CI)*	*p*	*BF*
Adult vs. child	0.308 (−0.416–1.031)	0.381	0.445
Infarct score change	−0.781 (−2.154–0.592)	0.245	0.424
mSS change	0.234 (−0.513–0.98)	0.517	0.366
WMD score change	1.931 (−0.496–4.359)	0.111	0.610
CVR score change	−0.036 (−0.371–0.298)	0.821	0.365
Other reason for cognitive defect	−0.474 (−2.260–1.312)	0.582	0.400
**Visuo-spatial functioning**	*B (95%CI)*	*p*	*BF*
Adult vs. child	0.321 (−0.298–0.94)	0.288	1.326
Infarct score change	0.317 (−0.820–1.454)	0.563	0.854
mSS change	−0.020 (−0.595–0.554)	0.941	0.668
WMD score change	2.11 (0.119–4.101)	**0.039**	2.205
CVR score change	−0.256 (−0.530–0.017)	0.064	1.632
Other reason for cognitive defect	−0.097 (−0.992–0.799)	0.822	0.677

*B*, unstandardized regression coefficient; *BF*, Bayes factor; *CI*, confidence interval; CVR, cerebrovascular reactivity; mSS, modified Suzuki score; WMD, white matter disease. Other reason for cognitive defects: patients with moyamoya syndrome with another condition influencing their cognition (e.g., Down’s syndrome).

## Data Availability

The data presented in this study are available on request from the corresponding author.

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
