# Peer review of "Clinical Outcome, Cognition, and Cerebrovascular Reactivity after Surgical Treatment for Moyamoya Vasculopathy: A Dutch Prospective, Single-Center Cohort Study"

_jcm, 2022, doi:10.3390/jcm11247427_

Round 1

Reviewer 1 Report

This is a nicely conducted study to assess the treatment effect of neurosurgical procedure of revascularization for Moyamoya disease.  This is apparently a follow up study by the same group published previously (PMID: 35523260).  The data was clearly presented and convincing.  The results and conclusions were reasonable. 

There were some differences between the adults and children, and the etiology of these differences can be discussed or speculated a bit more. 

Reviewer 2 Report

This is a prospective cohort study in a tertiary referral hospital, for 5 years (2012 to 2017), which analyzed the effect of surgical treatment of revascularization in children and adults with Moyamoya vasculopathy on clinical measures, including cognitive measures, functionality and quality of life, and the association between cognitive measures and cerebrovascular reactivity. The authors concluded that there was an improvement in cerebrovascular reactivity, in the frequency of transient ischemic attacks and in the measure of functionality in the subgroup of children, and headache frequency and quality of life in the subgroup of adults, after surgical treatment. There was stability of most cognitive scores, except for the improvement in language ability, evident in the subgroup of children. However, no association was observed between cerebrovascular reactivity and the cognitive measures evaluated.

The study presented is relevant because it presents multidimensional clinical data, related to compromised body structures and functions, functionality, and quality of life, on the effects of a procedure used in the medical treatment of people with Moyamoya vasculopathy, with an average longitudinal follow-up of 60 weeks.

However, there are some points that deserve the attention of the authors:

1. The main one refers to the need for discussion about the observed results regarding the function of language.  Language was the cognitive function with improvement in the group in the follow-up evaluation, however, this occurred specifically in the subgroup of children, with a mean age of 11 years (standard deviation 4.1). Considering the language acquisition and development process that takes place at this stage of life, the authors need to clarify how much of this improvement is due in fact to the effects of the intervention and not to the language development of these children.  It is possible to verify that the variation in the time between baseline and follow-up evaluation was very broad in this group (14 to 72 months), therefore, this possible bias needs to be discussed and/or pointed out as a limitation of the study, given the lack of a control group. It remains to be clarified whether these children underwent some type of speech therapy intervention in this period, considering that their baseline scores were apparently lower than expected, another factor that could have influenced the results obtained.

2. In the abstract, the objectives presented do not seem to be complete, considering the objectives presented in the manuscript body, since the investigation of the association between cognitive functions and cerebrovascular reactivity was not clearly presented.

3. Appendix I is shown as G.

4. Clarify whether the same set of tests was applied to all adults in the group who had the cognitive measures evaluated. This also applies to the subgroup of children. Apparently yes, but that was not clear to the reader.

5. In the discussion, page 14, lines 390-391, mentioned there were several studies, but none was referenced.

6. At the conclusion, page 15, lines 467-468, it would be appropriate to specify which cognitive improvement they refer to.
